nanotechnology

CoMoO$_4$ nanoplate array, layered hierarchical, hydrothermal, supercapacitor, areal capacitance

**Authors for correspondence:**
XiaoYan Hu
e-mail: 675960048@qq.com
Heng Wang
e-mail: wangheng8463@163.com

# Construction of layered hierarchical CoMoO$_4$ nanostructured arrays for supercapacitors with enhanced areal capacitance

## XiaoYan Hu[1], Hai Wang[2], SanMei Jin[2] and Heng Wang[1]

[1]School of New Energy and Electronic Engineering, Yancheng Teachers University, Yancheng 224007, People's Republic of China
[2]School of Mathematics and Physics, China University of Geosciences, Wuhan 430079, People's Republic of China

(iD) HW, 0000-0001-9723-6340

Layered hierarchical CoMoO$_4$ nano-structured arrays grown on nickel foam were designed and synthesized by a two-step hydrothermal method following by annealing. With the increase in the nearly three times loading mass of active materials, the specific capacitance of the layered hierarchical CoMoO$_4$ nano-structured arrays only shows a slight loss compared with the single-layer CoMoO$_4$ nano-structured arrays, which dramatically improved the areal capacitance from 2.47 to 6.79 F cm$^{-2}$. Also, the layered hierarchical CoMoO$_4$ nano-structured arrays showed 94.8% capacitance retention after 2500 cycles, which is mainly due to the well-designed layered hierarchical structure and good conductivity.

## 1. Introduction

Supercapacitors, with longer lifespan, better safety and faster charge–discharge capability, are being considered as most promising energy-storage devices [1–5]. Among the supercapacitor materials, transition metal oxides, CoO$_X$, NiO and MnO$_2$, have been widely investigated because of their high theoretical capacitance [6–8]. In recent years, ternary metal oxides, CoMoO$_4$, NiCo$_2$O$_4$ and MnMoO$_4$, have evoked tremendous interest for their better electrical conductivity and higher redox activity compared to single component [9,10]. CoMoO$_4$ is advantageous because of its low cost and non-toxic properties, and it also exhibits excellent electrochemical properties [11–13]. Wang and co-workers reported CoMoO$_4$ nanoplates on Ni foam which shows a remarkable specific capacitance of 2526 F g$^{-1}$ at 4 mA cm$^{-2}$. However, the

**Figure 1.** (*a*) Substrate nickel foam; (*b*) single-layer CoMoO$_4$ nanoplate arrays on the nickel foam; (*c*) layered hierarchical CoMoO$_4$ nanostructured arrays on the nickel foam; (*d*) electron and ion transfer in the layered hierarchical structure.

mass loading is about 0.5 mg cm$^{-2}$ and relatively low, the areal capacitance is 1.26 F cm$^{-2}$ [14]. In a practical application, high areal capacitance leads to higher energy density and output power supply, which is more significant. To further improve the CoMoO$_4$-related supercapacitor device and areal capacitance, Liu and co-workers [15,16] synthesized hierarchical CoMoO$_4$@NiMoO$_4$ core–shell nanosheet arrays and CoMoO$_4$@MnO$_2$ core–shell structure on nickel foam, which had a high areal capacitance of 3.3 F cm$^{-2}$ and 2.27 F cm$^{-2}$, respectively.

As seen, core–shell structure could lead to an improved specific capacitance and electrochemical behaviours. However, the shell is usually very thin and the low mass loading still limits the practical application. When the thickness is too great, the shell would in turn restrain the ion transfer to the inner core. Herein, a novel layered hierarchical nano-structured CoMoO$_4$ on the three-dimensional nickel foam is designed to get high areal capacitance. The first layer is designed as CoMoO$_4$ nanoplate arrays and the second layer is designed as nanoflower-like structure deposited on the first layer. Such a layered hierarchical structure possesses the following advantages: (i) CoMoO$_4$ nanoplate arrays on the nickel foam have a high electrochemical activity and abundant redox sites, and, as ternary metal oxide nanoarrays, CoMoO$_4$ nanoplate arrays can provide direct charge transfer channel for the second layer; (ii) the second layer CoMoO$_4$ nanoflower structure on the CoMoO$_4$ nanoplate arrays is not closely packed and electrolyte could penetrate into the first layer directly; (iii) a high mass loading could be facilely achieved for this structure and the areal capacitance is improved. As a result, the designed layered hierarchical nano-structured CoMoO$_4$ on the three-dimensional nickel foam exhibits a high areal capacitance of 6.79 F cm$^{-2}$ at 5 mA cm$^{-2}$, which is much higher than single-layer CoMoO$_4$ nanoplate arrays. Also, at a high current density of 50 mA cm$^{-2}$, the layered hierarchical nano-structures still have 94.8% capacitance retention after 2500 cycles, which implies that our structure design is efficient to improve the areal capacitance and can be applied to construct the structure of other electrode materials.

# 2. Experimental and computational section

All the reagents were purchased from Sinopharm Chemical Reagent Co. Ltd; Co(NO$_3$)$_2$·6H$_2$O, 98%; Na$_2$MoO$_4$·7H$_2$O, 98%. The thickness of the nickel foam is about 1.6 mm and the porosity is 96.7%, 110 PPI. The brief fabrication procedure is illustrated in figure 1. The detail is given below.

## 2.1. Synthesis of single-layer CoMoO$_4$ nanoplate arrays on nickel foam

The substrate nickel foam was rinsed with ethanol, deionized water and hydrochloric acid solution. To obtain the CoMoO$_4$ nanoplate arrays on the nickel foam, 2.5 mmol Co(NO$_3$)$_2$·6H$_2$O and 2.5 mmol NaMoO$_4$·7H$_2$O were mixed and stirred for 10 min, then the solution was transferred into a Teflon-lined stainless steel autoclave liner with the PTFE tape wrapped nickel foam immersed. The liner was sealed in a Teflon-lined stainless steel autoclave and maintained in an electric oven at 180°C for 12 h. After the reaction, the nickel foam with the arrays were rinsed with deionized water several times and dried at 60°C for 5 h. Finally, the sample was heated at 300°C for 2 h and single-layer CoMoO$_4$

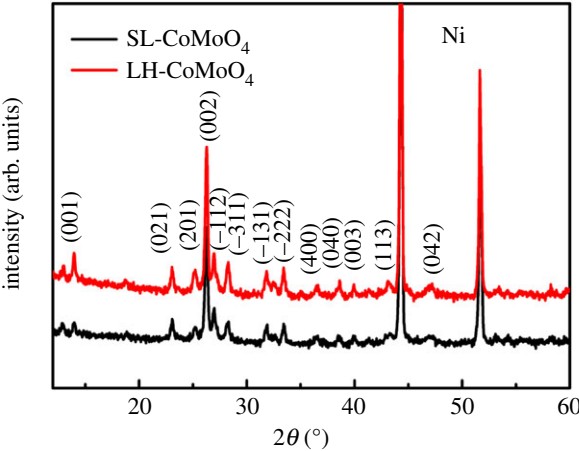

**Figure 2.** XRD patterns of the SL-CoMoO₄ and LH-CoMoO₄ on the nickel foam.

nanoplate arrays on the nickel foam was obtained. The weight of the active material was 1.85 mg cm⁻². This structure is denoted as SL-CoMoO₄.

## 2.2. Synthesis of layered hierarchical CoMoO₄ nanostructured arrays on the nickel foam

To obtain the layered hierarchical nanostructured array on the nickel foam, the obtained single-layer CoMoO₄ nanoplate arrays on the nickel foam were put into the same solution as described above. The same hydrothermal reaction was repeated at 180°C for 9 h. After the reaction, the nickel foam with the layered structures were rinsed with deionized water several times and dried at 60°C for 5 h. Finally, the sample was heated at 300°C for 2 h and layered hierarchical CoMoO₄ nanostructured arrays on the nickel foam were obtained. The weight of the active material was 4.95 mg cm⁻². This structure is denoted as LH-CoMoO₄.

## 2.3. Characterization methods

The products were characterized using X-ray diffraction (XRD, PANalytical Empyrean, Cu Kα radiation; $\lambda = 1.5418$ Å) and field-emission scanning electron microscopy (FESEM, JEOL JSM-6700F, 10 kV); the mass of the electrode materials was measured on an AX/MX/UMX balance (METTLER TOLEDO, maximum = 5.1 g; $d = 0.001$ mg). To characterize the electrochemical behaviours, CHI 660D (CH Instruments Inc., Shanghai) electrochemical workstation was used in a three-electrode electrochemical cell using a 6 M KOH aqueous solution as electrolyte. Electrochemical impedance spectroscopy was tested by applying an AC voltage with 5 mV amplitude in a frequency range from 100 kHz to 0.1 Hz at open circuit potential. To test the electrochemical behaviours of LH-CoMoO₄ electrodes, $2 \times 0.5$ cm² nickel foam with the active material was cut as the working electrode, Ag/AgCl electrode was used as the reference electrode and Pt foil was used as the counter electrode. The electrolyte is 6 M KOH aqueous solution. SL-CoMoO₄ was also tested for comparison.

## 3. Results and discussion

XRD patterns of the SL-CoMoO₄ and the LH-CoMoO₄ on the nickel foam are displayed in figure 2. As shown, for the single-layer CoMoO₄, two ultrahigh peaks come from the substrate Ni, other peaks can be indexed to the (001), (021), (201), (002), (−112), (−311), (−131), (−222), (400), (040), (003), (113), (042) lattice planes, which corresponds to the monoclinic CoMoO₄ (JCPDS card No. 21-0868). After second hydrothermal reaction for 9 h, XRD spectrum of LH-CoMoO₄ shows no obvious difference from the SL-CoMoO₄, which indicates product obtained during the second step is also CoMoO₄.

The scanning electron microscope (SEM) image of SL-CoMoO₄ on the nickel foam is shown in figure 3a,b. Figure 3a displays the top view of SL-CoMoO₄; obviously, the nanoplate arrays grew vertically and uniformly on the nickel foam. The cross-section of SL-CoMoO₄ is shown in figure 3b. The thickness of the array is about 1.5 µm. Figure 3c−f is the SEM images of LH-CoMoO₄. As shown in figure 3c,d, the second layer CoMoO₄ is mostly constituted of CoMoO₄ nanoflowers. The diameter

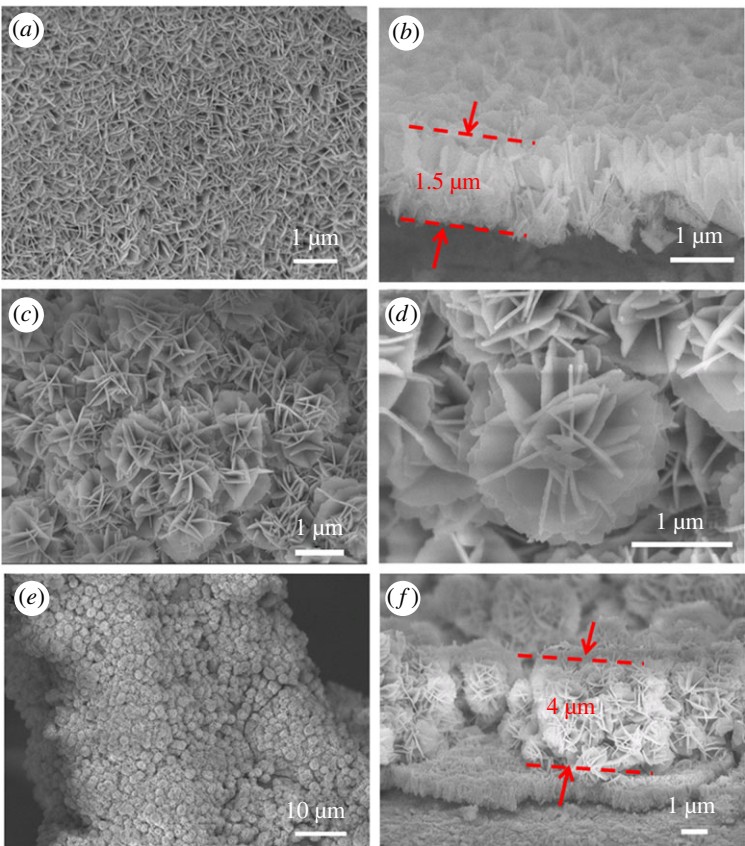

**Figure 3.** SEM images: (*a*) top view image of SL-CoMoO₄; (*b*) cross-section image of the SL-CoMoO₄; (*c*) top view image of LH-CoMoO₄; (*d*) high magnification of single nanoflower from LH-CoMoO₄; (*e*) low magnification of LH-CoMoO₄; (*f*) cross-section image of LH-CoMoO₄.

of the $CoMoO_4$ nanoflowers is about $1-2$ μm and the nanoflower is also made up of randomly oriented $CoMoO_4$ nanoplates. Figure 3*e* displays low magnification of LH-$CoMoO_4$ on single nickel wire on the nickel foam, which indicates that the first layer nanoplate arrays are covered with $CoMoO_4$ nanoflowers. Figure 3*f* displays the cross-section image of LH-$CoMoO_4$ on the nickel foam. Obviously, the nanoflowers closely contact with each other and are directly deposited on the nanoplate array layer. The thickness is about 4 μm. It was worth noting that the $CoMoO_4$ nanoflower-like structure did not completely cover the first layer because some gaps existed among the nanoflower structures. Thus, the electrolyte can penetrate into the first layer directly, which implies that our layered nanoflower structure would not restrain the ion transport on the basis of increased mass loading.

From the XRD, SEM and other related works, we can conclude the growth process of our LH-$CoMoO_4$ as follows: at the first hydrothermal reaction stage, $Co^{2+}$ and $MoO_4^{2-}$ ions combined and formed nanoparticles as the seeds on the three-dimensional nickel foam at a supersaturated solution. After that, the nanoparticle grew in a preferred crystal growth direction and formed vertically oriented nanoplate arrays. At the second hydrothermal reaction stage, because the first layer has grown on the substrate, there is no flat surface for the growth of the seed. The seed directly nucleated in the solution and grew up into nanoplate randomly, formed the nanoflowers [10,17]. Finally, the nanoflowers deposited on the first layer of $CoMoO_4$ and LH-$CoMoO_4$ was obtained.

Figure 4*a* displays the cyclic voltammetry (CV) of SL-$CoMoO_4$ and LH-$CoMoO_4$ at 5 mV s⁻¹. Redox peaks correspond to the redox reactions of $Co^{2+}/Co^{3+}$ and $Co^{3+}/Co^{4+}$. Obviously, the LH-$CoMoO_4$ electrode has a higher CV loop area than SL-$CoMoO_4$, which shows a better capacitive behaviour. At a current density of 5 mA cm⁻², LH-$CoMoO_4$ can discharge for 761.4 s, which is much higher than that of SL-$CoMoO_4$, as shown in figure 4*b*. Figure 4*c* displays the CV test of LH-$CoMoO_4$ at different scanning rates in the potential window $-0.2-0.6$ V. Areal capacitance of SL-$CoMoO_4$ and LH-$CoMoO_4$ at different current densities are shown in figure 4*d*. At a current density of 5 mA cm⁻², LH-$CoMoO_4$ exhibits high areal capacitance of 6.79 F cm⁻², which is much higher than the SL-$CoMoO_4$ of 2.47 F cm⁻². It is also higher than other $CoMoO_4$-related work (such as

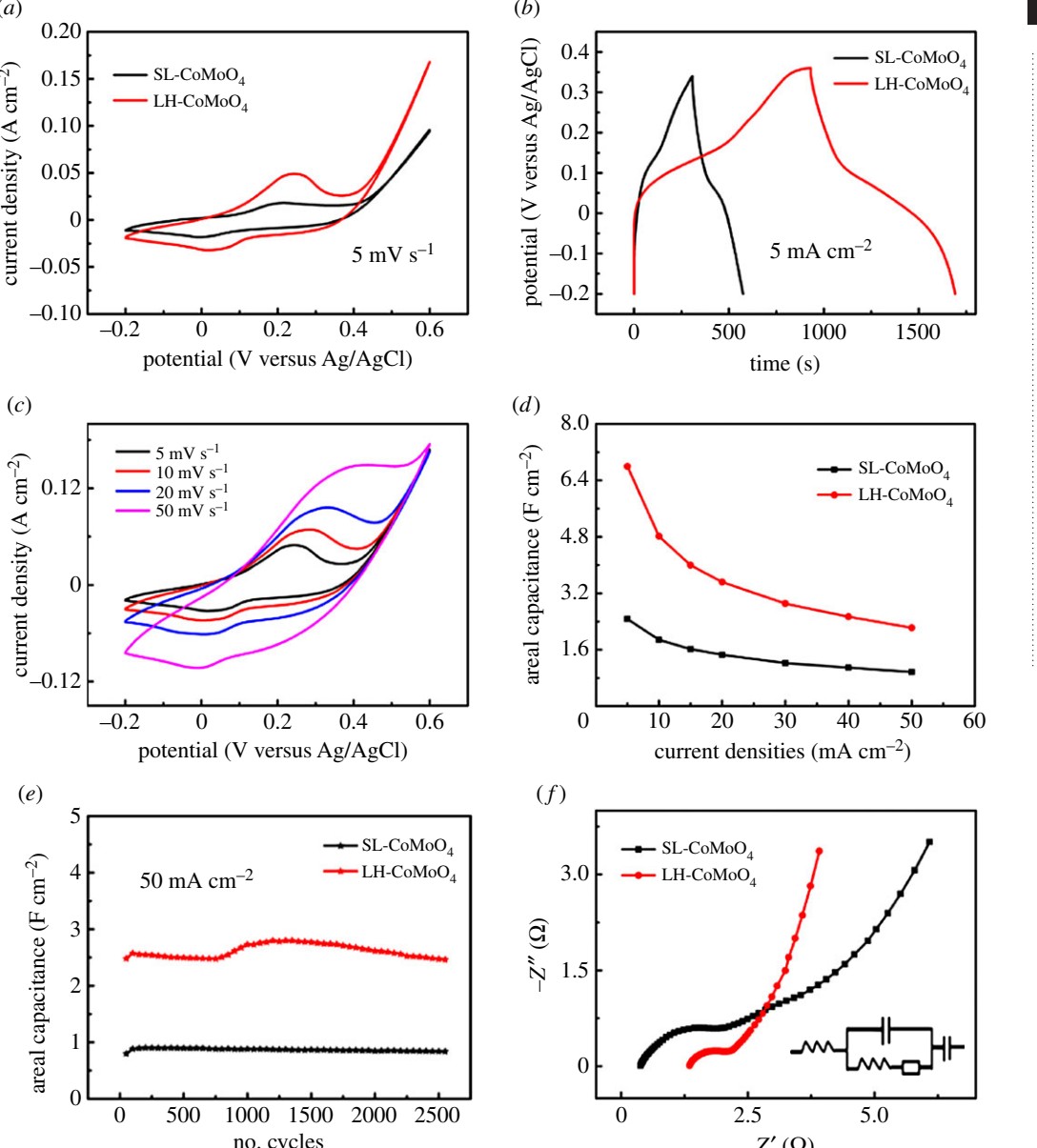

**Figure 4.** Electrochemical test of SL-CoMoO$_4$ and LH-CoMoO$_4$: (*a*) CV comparison at 5 mV s$^{-1}$; (*b*) charge–discharge curves at 5 mA cm$^{-2}$; (*c*) CV test of LH-CoMoO$_4$ at different scanning rates; (*d*) areal capacitance at different current densities; (*e*) cycle performance at 50 mA cm$^{-2}$; (*f*) Nyquist plots.

CoMoO$_4$@NiMoO$_4$ core–shell nanosheet arrays on nickel foam, 3.3 F cm$^{-2}$ at 8 mA cm$^{-2}$; CoMoO$_4$@MnO$_2$ core–shell structure aligned on Ni foam, 2.27 F cm$^{-2}$ at 3 mA cm$^{-2}$ [15,16]).

Meanwhile, the mass of the electrode materials loading on the nickel foam was measured on a balance. The loading mass was calculated to be 1.85 and 4.95 mg cm$^{-2}$ for LH-CoMoO$_4$ and SL-CoMoO$_4$, respectively. Considering the loading mass, the specific capacitances of LH-CoMoO$_4$ and SL-CoMoO$_4$ are 1331.3 and 1372.2 F g$^{-1}$, respectively. Although the mass loading has increased three times, specific capacitance only shows a slight loss. Even at a high rate of 50 mA cm$^{-2}$, LH-CoMoO$_4$ still remained areal capacitance of 2.22 F cm$^{-2}$, showing good rate capability. The cycle tests of the SL-CoMoO$_4$ and LH-CoMoO$_4$ at 50 mA cm$^{-2}$ are shown in figure 4*e*. Both SL-CoMoO$_4$ and LH-CoMoO$_4$ show excellent cycle performance. After 2500 cycles, LH-CoMoO$_4$ still retains 94.8% capacitance of the initial capacitance. Especially deserving to be mentioned, after 1000 cycles, there even exists a rising cycling performance which can be ascribed to the layered hierarchical structure and the compact but porous nanostructure: (i) the bottom of the second layer is closely connected to the top of the first layer with the reaction progressing, so that the structure damage caused by volume expansion during

the cycling process was alleviated, resulting in enhanced stability. (ii) The compact but porous nanostructure also helps to alleviate the structure damage caused by volume expansion during the cycle test. Figure 4*f* shows the Nyquist plots with the equivalent circuit in the inset of SL-CoMoO$_4$ and LH-CoMoO$_4$, respectively. The diameter of the semicircle in the mid-frequency is the charge-transfer resistance ($R_{ct}$) of the electrode material. For SL-CoMoO$_4$ and LH-CoMoO$_4$, the $R_{ct}$ values are 2.32 and 1.18 $\Omega$, which indicates that the LH-CoMoO$_4$ still had high charge conductivity. These results convince that the LH-CoMoO$_4$ is a very promising supercapacitor electrode material and the layered hierarchical nanostructure is an efficient structure to improve the areal capacitance.

# 4. Conclusion

A layered hierarchical nano-structured CoMoO$_4$ grown on nickel foam has been designed by a facile two-step hydrothermal reaction following annealing treatment. The LH-CoMoO$_4$ shows a high areal capacitance of 6.79 F cm$^{-2}$, which is much higher than the SL-CoMoO$_4$ of 2.47 F cm$^{-2}$. Meanwhile, the LH-CoMoO$_4$ shows good conductivity and excellent cycle performance, which is mainly due to the well-designed nanostructure. Such outstanding electrochemical behaviours confirm that our structure design is efficient to improve the areal capacitance and can be applied to construct the structure of other electrode materials.

Data accessibility. There are no additional data to accompany this manuscript. All relevant datasets are within the main body of the manuscript. All protocols and software used are stated fully in the methodology section. There are no coding lists for this research.

Authors' contributions. X.Y.H. and H.W. participated in the main experimental work. S.M.J. and H.W. carried out the data analysis and drafted the manuscript. All the authors gave their final approval of the version for publication.

Competing interests. We have no competing interests.

Funding. This work was financially supported by the National Natural Science Foundation of China (no. 11647073).

Acknowledgements. We thank all the staff in Yancheng Teachers University for meaningful discussions of the research.

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
