## [Reviewer comments · Royal Society Open Science]

Review History

RSOS-181592.R0 (Original submission)

Review form: Reviewer 1

Is the manuscript scientifically sound in its present form?

Yes

Are the interpretations and conclusions justified by the results?

Yes

Is the language acceptable?

No

Is it clear how to access all supporting data?

Not Applicable

Do you have any ethical concerns with this paper?

No

Have you any concerns about statistical analyses in this paper?

I do not feel qualified to assess the statistics

Recommendation?

Major revision is needed (please make suggestions in comments)

Comments to the Author(s)

The paper contains original material but is written unclearly. Overall, I think the paper can be accepted for publication in Royal Society Open Science if the authors should clarify the following unclear questions.

- 1) What's the thickness of the CoMoO₄ nanoflower-like structures? Is there any relationship between the thickness and capacitance? I read the manuscript several times and found it very preliminary with little data. To optimize synthesis condition, I strongly suggest the authors make the performance comparison among more samples.
- 2) The title is unclear, and the English in text should be finely polished.

Review form: Reviewer 2

Is the manuscript scientifically sound in its present form?

Yes

Are the interpretations and conclusions justified by the results?

Yes

Is the language acceptable?

Yes

Is it clear how to access all supporting data?

Yes

Do you have any ethical concerns with this paper?

No

Have you any concerns about statistical analyses in this paper?

I do not feel qualified to assess the statistics

Recommendation?

Accept with minor revision (please list in comments)

Comments to the Author(s)

Hu et al described the fabrication of a layered hierarchical nano-structured CoMoO₄ on nickel foam by a facile two-step hydrothermal reaction following annealing treatment and their applications for supercapacitor. The LH-CoMoO₄ shows a high areal capacitance of 6.79 F cm⁻², which is much higher than the SL-CoMoO₄ of 2.47 F cm⁻². The nanostructured LH-CoMoO₄ exhibits good conductivity and excellent cycle performance, which confirmed the importance of nanostructures reported herein. This work affords an emerging and facile method to obtain advanced energy materials. The reason for the excellent performance is also demonstrated. This work is a very important contribution to Royal Society Open Science. It can be appeared as a full paper after following revisions:

1. What is the areal loading of LH-CoMoO₄? How they anchoring onto the Ni foam?

2. In Figure 3, 'um' should be updated as ' $\mu\text{ m}$ '.
3. Figure 4e exhibits the cycle performance at 50 mA cm⁻². The areal density is divided by the surface of electrode or the active materials?
4. The recent advances in nanostructured electrode for supercapacitor application (e.g. CoMoO₄/Co_{1-x}S hybrid on Ni foam for high-performance supercapacitors reported in J Energy Chem 2018, 2, 478-485; CoMoO₄-3D graphene hybrid electrodes reported in Adv Mater 2013, 26, 1044-1051; Hollow CoMoO₄ spheres for supercapacitors reported in Chem Commun 2018, 54, 10355-10358) should be introduced in the revised backgrounds.

Decision letter (RSOS-181592.R0)

13-Nov-2018

Dear Dr Wang:

Title: Construction of Layered Hierarchical CoMoO₄ Nanostructured Arrays With Enhanced Areal Capacitance

Manuscript ID: RSOS-181592

Thank you for submitting the above manuscript to Royal Society Open Science. On behalf of the Editors and the Royal Society of Chemistry, I am pleased to inform you that your manuscript will be accepted for publication in Royal Society Open Science subject to minor revision in accordance with the referee suggestions. Please find the reviewers' comments at the end of this email.

The reviewers and handling editors have recommended publication, but also suggest some minor revisions to your manuscript. Therefore, I invite you to respond to the comments and revise your manuscript.

Please also include the following statements alongside the other end statements. As we cannot publish your manuscript without these end statements included, if you feel that a given heading is not relevant to your paper, please nevertheless include the heading and explicitly state that it is not relevant to your work. We have included a screenshot example of the end statements for reference.

- Ethics statement

Please clarify whether you received ethical approval from a local ethics committee to carry out your study. If so please include details of this, including the name of the committee that gave consent in a Research Ethics section after your main text. Please also clarify whether you received informed consent for the participants to participate in the study and state this in your Research Ethics section.

OR

Please clarify whether you obtained the necessary licences and approvals from your institutional animal ethics committee before conducting your research. Please provide details of these licences and approvals in an Animal Ethics section after your main text.

OR

Please clarify whether you obtained the appropriate permissions and licences to conduct the fieldwork detailed in your study. Please provide details of these in your methods section.

Because the schedule for publication is very tight, it is a condition of publication that you submit the revised version of your manuscript before 22-Nov-2018. Please note that the revision

deadline will expire at 00.00am on this date. If you do not think you will be able to meet this date please let me know immediately.

Best wishes,
Dr Laura Smith
Publishing Editor, Journals

On behalf of the Subject Editor Professor Anthony Stace and the Associate Editor Professor Eva Hevia.

RSC Associate Editor:
Comments to the Author:
(There are no comments.)

RSC Subject Editor:
Comments to the Author:
(There are no comments.)

Reviewer comments to Author:
Reviewer: 1

Comments to the Author(s)

The paper contains original material but is written unclearly. Overall, I think the paper can be accepted for publication in Royal Society Open Science if the authors should clarify the following unclear questions.

- 1) What's the thickness of the CoMoO₄ nanoflower-like structures? Is there any relationship between the thickness and capacitance? I read the manuscript several times and found it very preliminary with little data. To optimize synthesis condition, I strongly suggest the authors make the performance comparison among more samples.
- 2) The title is unclear, and the English in text should be finely polished.

Reviewer: 2

Comments to the Author(s)

Hu et al described the fabrication of a layered hierarchical nano-structured CoMoO₄ on nickel foam by a facile two-step hydrothermal reaction following annealing treatment and their applications for supercapacitor. The LH-CoMoO₄ shows a high areal capacitance of 6.79 F cm⁻², which is much higher than the SL-CoMoO₄ of 2.47 F cm⁻². The nanostructured LH-CoMoO₄ exhibits good conductivity and excellent cycle performance, which confirmed the importance of nanostructures reported herein. This work affords an emerging and facile method to obtain advanced energy materials. The reason for the excellent performance is also demonstrated. This work is a very important contribution to Royal Society Open Science. It can be appeared as a full paper after following revisions:

1. What is the areal loading of LH-CoMoO₄? How they anchoring onto the Ni foam?
2. In Figure 3, 'um' should be updated as 'μ m'.
3. Figure 4e exhibits the cycle performance at 50 mA cm⁻². The areal density is divided by the surface of electrode or the active materials?
4. The recent advances in nanostructured electrode for supercapacitor application (e.g. CoMoO₄/Co_{1-x}S hybrid on Ni foam for high-performance supercapacitors reported in J Energy Chem 2018, 2, 478-485; CoMoO₄-3D graphene hybrid electrodes reported in Adv Mater 2013, 26, 1044-1051; Hollow CoMoO₄ spheres for supercapacitors reported in Chem Commun 2018, 54, 10355-10358) should be introduced in the revised backgrounds.

Author's Response to Decision Letter for (RSOS-181592.R0)

See Appendix A.

Decision letter (RSOS-181592.R1)

18-Dec-2018

Dear Dr Wang:

Title: Construction of Layered Hierarchical CoMoO₄ Nanostructured Arrays for Supercapacitors with Enhanced Areal Capacitance

Manuscript ID: RSOS-181592.R1

It is a pleasure to accept your manuscript in its current form for publication in Royal Society Open Science. The chemistry content of Royal Society Open Science is published in collaboration with the Royal Society of Chemistry.

The comments of the reviewer(s) who reviewed your manuscript are included at the end of this email. I apologise that this took longer than usual.

On behalf of the Subject Editor Professor Anthony Stace and the Associate Editor Professor Eva Hevia.

RSC Associate Editor
Comments to the Author:
(There are no comments.)

Reviewer(s)' Comments to Author:

Appendix A

Dear editor and reviewers,

I am very pleased to hear from you, thank you very much for your comments to my manuscript! According to your comments, a revision is made for my paper, corrected sentences and words are marked with red color in the revised paper. Some explanations are shown as follow:

REVIEWER REPORT(S):

Reviewer: 1

Comments:

The paper contains original material but is written unclearly. Overall, I think the paper can be accepted for publication in Royal Society Open Science if the authors should clarify the following unclear questions.

Reply: We appreciate the reviewer for the encouraging comments, we hope the revised manuscript can be more valuable and interesting to the readers of *Royal Society Open Science*.

1. What's the thickness of the CoMoO₄ nanoflower-like structures? Is there any relationship between the thickness and capacitance? I read the manuscript several times and found it very preliminary with little data. To optimize synthesis condition, I strongly suggest the authors make the performance comparison among more samples.

Reply: We have marked the thickness of the CoMoO₄ nanoflower-like structures in Fig. 3f, and the thickness is about 4 μm. Yes, the capacitive behavior of electrode can be effected by the thickness.

However, in this work, we only investigated the effects the loading mass of the layered hierarchical CoMoO₄ nano-structured arrays on the areal capacitance.

2. The title is unclear, and the English in text should be finely polished.

Reply: We agree with the reviewer's comments. We have modified the title and the English with red color in the revised paper.

Reviewer: 2

Comments:

Hu et al described the fabrication of a layered hierarchical nano-structured CoMoO₄ on nickel foam by a facile two-step hydrothermal reaction following annealing treatment and their applications for supercapacitor. The LH-CoMoO₄ shows a high areal capacitance of 6.79 F cm⁻², which is much higher than the SL-CoMoO₄ of 2.47 F cm⁻². The nanostructured LH-CoMoO₄ exhibits good conductivity and excellent cycle performance, which confirmed the importance of nanostructures reported herein. This work affords an emerging and facile method to obtain advanced energy materials. The reason for the excellent performance is also demonstrated. This work is a very important contribution to Royal Society Open Science. It can be appeared as a full paper after following revisions:

Reply: We appreciate the reviewer for the encouraging comments, we hope the revised manuscript can be more valuable and interesting to

the readers of *Royal Society Open Science*.

1. What is the areal loading of LH-CoMoO₄? How they anchoring onto the Ni foam?

Reply: The areal loading of LH-CoMoO₄ is 4.95 mg cm⁻². The layered hierarchical CoMoO₄ nanostructured on the array is formed via heterogeneous nucleation in the solution. After that, the nanostructured CoMoO₄ were deposited on the array by surface absorption.

2. In Figure 3, 'um' should be updated as 'μm'.

Reply: Thanks for the reviewer's remanding. We have updated 'um' with 'μm' in Figure 3.

3. Figure 4e exhibits the cycle performance at 50 mA cm⁻². The areal density is divided by the surface of electrode or the active materials?

Reply: The areal density is divided by the surface of electrode, which is also the active materials. This is because the surface of electrode is fully covered by the active materials.

4. The recent advances in nanostructured electrode for supercapacitor application (e.g. CoMoO₄/Co_{1-x}S hybrid on Ni foam for high-performance supercapacitors reported in J Energy Chem 2018, 2, 478-485; CoMoO₄-3D graphene hybrid electrodes reported in Adv Mater 2013, 26, 1044-1051; Hollow CoMoO₄ spheres for supercapacitors reported in Chem Commun 2018, 54, 10355-10358) should be introduced in the revised backgrounds.

Reply: Thanks for the reviewer's remanding. The papers mentioned by the reviewer are excellent works, and we have supplemented these

references, please see references 11, 12 and 13.

The manuscript has been revised according to your comments. We hope the revised paper would satisfy you and the reviewers. If there are any problems, please do not hesitate to contact me.

We are looking forward to hearing from you soon.

Sincerely yours,

Xiaoyan Hu